# The role of PM2.5 exposure in lung cancer: mechanisms, genetic factors, and clinical implications

Chi-Yuan Chen[1,2,12], Kuo-Yen Huang [ID] [3,4,5,12], Chin-Chuan Chen[2,6], Ya-Hsuan Chang[7,8], Hsin-Jung Li[5], Tong-Hong Wang [ID] [1,2,6,9 ✉] & Pan-Chyr Yang [ID] [5,10,11 ✉]

## Abstract

**Lung cancer is one of the most critical global health threats, as the second most common cancer and leading cause of cancer deaths globally. While smoking is the primary risk factor, an increasing number of cases occur in nonsmokers, with lung cancer in non-smokers (LCNS) now recognized as the fifth leading cause of cancer mortality worldwide. Recent evidence identifies air pollution, particularly fine particulate matter (PM2.5), as a significant risk factor in LCNS. PM2.5 can increase oxidative stress and inflammation, induce genetic alterations and activation of onco-genes (including the epidermal growth factor receptor, EGFR), and contribute to lung cancer progression. This review summarizes the current understanding of how exposure to PM2.5 induces lung carcinogenesis and accelerates lung cancer development. It underscores the importance of prevention and early detection while calling for targeted therapies to combat the detrimental effects of air pollution. An integrated approach that combines research, public health policy, and clinical practice is essential to reduce the lung cancer burden and improve outcomes for those affected by PM2.5 exposurrre.**

**Keywords** PM2.5; Inflammation; DNA Damage; EGFR; Lung Cancer
**Subject Categories** Cancer; Evolution & Ecology; Respiratory System

## Introduction

Lung cancer is a significant global health threat, being the second most common malignancy and the leading cause of cancer-related death worldwide. It accounts for more than 2.2 million new cases and 1.8 million deaths annually (Sung et al, 2021). Tobacco smoking remains the main risk factor for lung cancer and is responsible for two-thirds of all lung cancer deaths (Sung et al, 2021). However, it is estimated that at least one-third of lung cancer cases occur in nonsmokers (LoPiccolo et al, 2024), and its incidence has steadily increased over time (Jeon et al, 2018). When considered independently, lung cancer in nonsmokers (LCNS) is the fifth most common cause of cancer-related death (Fig. 1) (LoPiccolo et al, 2024).

Air pollutants, especially fine particulate matter (PM2.5, particulate matter with aerodynamic diameter of ≤2.5 μm), play a significant role in lung cancer progression, driver mutation activation, and treatment resistance (Colín-Val et al, 2024; Hill et al, 2023). Long-term exposure to PM2.5 has been shown to increase lung cancer risk, particularly in individuals with high genetic susceptibility (Huang et al, 2021). Recent studies have illuminated the intricate relationship between PM2.5 exposure and LCNS, with a focus on the prevalence and impact of driver mutations, especially in the epidermal growth factor receptor (EGFR) mutation (Adib et al, 2022). PM2.5-induced oxidative stress and inflammation have been implicated in driving the occurrence of EGFR driver mutations in lung cells (Hill et al, 2023), thus fueling the malignant transformation process. Moreover, PM2.5 exposure may potentiate the acquisition of genetic alterations that confer resistance to conventional therapies, posing formidable challenges in the management of LCNS. The elucidation of these potential mechanisms underscores the critical importance of proactive measures aimed at the prevention and early detection of lung cancer, particularly among nonsmokers.

In light of these pressing concerns, a comprehensive review exploring the interplay between LCNS and air pollution, with a specific focus on PM2.5, is warranted. This review begins with a brief description of PM2.5 and reviews the epidemiological evidence and possible mechanisms underlying its role in LCNS, including EGFR mutation, inflammation, and resistance to EGFR–tyrosine kinase inhibitors (TKIs). We aim to consolidate existing knowledge, identify gaps in understanding, and advocate

[1]Graduate Institute of Health Industry Technology and Research Center for Food and Cosmetic Safety, Chang Gung University of Science and Technology, Taoyuan, Taiwan. [2]BioBank, Chang Gung Memorial Hospital at Linkou, Taoyuan, Taiwan. [3]Department of Clinical Laboratory Sciences and Medical Biotechnology, College of Medicine, National Taiwan University, Taipei, Taiwan. [4]Program for Precision Health and Intelligent Medicine, Graduate School of Advanced Technology, National Taiwan University, Taipei, Taiwan. [5]National Taiwan University YongLin Institute of Health, National Taiwan University, Taipei, Taiwan. [6]Graduate Institute of Natural Products, Chang Gung University, Taoyuan, Taiwan. [7]Institute of Molecular and Genomic Medicine, National Health Research Institutes, Miaoli, Taiwan. [8]Research Center for Precision Environmental Medicine, Kaohsiung Medical University, Kaohsiung, Taiwan. [9]Department of Hepato-Gastroenterology, Liver Research Center, Chang Gung Memorial Hospital at Linkou, Taoyuan, Taiwan. [10]Institute of Biomedical Sciences, Academia Sinica, Taipei, Taiwan. [11]Department of Internal Medicine, National Taiwan University Hospital and National Taiwan University College of Medicine, Taipei, Taiwan. [12]These authors contributed equally: Chi-Yuan Chen, Kuo-Yen Huang. ✉E-mail: cellww@cgmh.org.tw; pcyang@ntu.edu.tw

**Glossary**

| | |
|---|---|
| **Adenocarcinoma** | A type of cancer that forms in mucus-secreting glands and is the most common type of NSCLC. |
| **Air pollution** | The presence of harmful substances in the air, including particulate matter, gases, and biological molecules, that can pose health risks to humans and the environment. |
| **Epidermal growth factor receptor (EGFR)** | A transmembrane protein that, when mutated, is key to the initiation and progression of cancers like lung cancer. |
| **EGFR–tyrosine kinase inhibitors (EGFR–TKIs)** | Targeted cancer therapies that block the action of EGFR and are used in treating cancers with EGFR mutations. |
| **Lung cancer in nonsmokers (LCNS)** | Lung cancer occurs in individuals who have never smoked or have smoked fewer than 100 cigarettes in their lifetimes. |
| **Oxidative stress** | A condition where excess ROS lead to cellular damage, contributing to cancer and other diseases. |
| **Particulate matter (PM)** | A mixture of tiny solid particles and liquid droplets suspended in the atmosphere. PM2.5 and PM10 refer to particles small enough to enter the respiratory system, with finer particles posing greater health risks. |
| **PM2.5** | Particulate matter 2.5 micrometers or smaller can reach deep into the lungs and enter the bloodstream, potentially causing numerous health problems. |
| **Reactive oxygen species (ROS)** | Oxygen-containing reactive molecules that, in excessive, can damage DNA, proteins, and cell membranes, contributing to diseases, including cancer. |

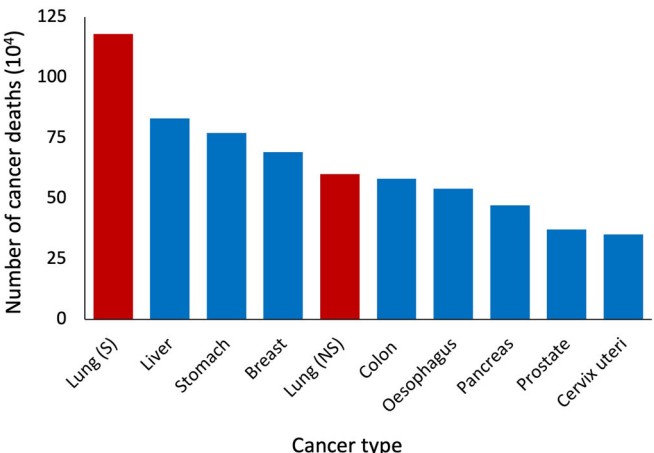

**Figure 1. Lung cancer in smokers and nonsmokers among the top 10 leading causes of cancer death worldwide in 2020.**

Lung cancer in smokers (S) and nonsmokers (NS), highlighted in red, represents the first and fifth leading causes of cancer death, respectively. Modified from (LoPiccolo et al, 2024).

for evidence-based interventions to mitigate the adverse impact of air pollution on lung health.

## PM2.5 in air pollution

Air pollution refers to the contamination of both indoor and outdoor environments by various harmful agents, including chemical, physical, and biological substances. It poses one of the most severe environmental threats to human health globally. The World Health Organization (WHO) estimates that the ambient and household air pollution together contribute to around 7 million premature deaths annually (Velasco and Jarosińska, 2022). This alarming statistic highlights the urgent need for effective measures to reduce air pollution and mitigate its detrimental impacts on human health.

In most countries, monitoring systems are in place to measure the levels of environmental pollutants such as particulate matter (PM), ozone, nitrogen dioxide, and sulfur dioxide (Loomis et al, 2013). PM is regarded as a key risk factor for cancer and was classified by the International Agency for Research on Cancer (IARC) as a Group 1 carcinogen in 2013 (Loomis et al, 2013, 2014). PM consists of airborne solid particles and liquid droplets that can enter the human body through breathing. PM has a complex composition that includes chemical components such as nitrates, sulfates, ammonium, elemental carbon, heavy metals, and polycyclic aromatic hydrocarbons (PAHs) (Sicard et al, 2019).

PM with larger particle sizes is typically blocked by the nasal cavity and upper respiratory tract, whereas particles with a diameter of ≤10 µm (PM10 and PM2.5) can penetrate deeply into the small airways and alveoli of the lungs and even enter the bloodstream, thereby causing damage to the respiratory and cardiovascular systems (Sicard et al, 2019). Compared with PM10, PM2.5 in particular has a greater tendency to adsorb toxic heavy metals (e.g., zinc, lead, arsenic, and cadmium) and organic pollutants (e.g., PAHs), and it possesses a stronger penetrating ability. Upon entry into the body, these toxic substances promote the release of reactive oxygen species (ROS) by macrophages. The oxidative stress induced by ROS leads to DNA damage and tissue inflammation, which trigger abnormal gene expression and increase the risk of cancer development (Thangavel et al, 2022; Valavanidis et al, 2013). In addition to PM2.5, a number of air pollutants also contribute significantly to cancer development (Turner et al, 2020). While pollutants like nitrogen oxides (NOx), ozone ($O_3$), and volatile organic compounds (VOCs) have been associated with lung cancer (Kurt et al, 2016), evidence points to PM2.5 as the most critical factor (Desikan, 2017). Its small size and complex chemical makeup give PM2.5 a unique ability to impact biological systems. Unlike NOx or $SO_2$, which primarily cause respiratory irritation and inflammation, PM2.5 penetrates deeper into lung tissues and persists longer, allowing its carcinogenic components to interact with lung cells more directly and over extended periods (Thangavel et al, 2022).

In developed countries, where industrial activity and vehicle emissions are high, air pollutants like PM2.5, $NO_2$, and ozone are often significant contributors to lung cancer. On the other hand,

reliance on coal and biomass for fuel in developing countries exposes populations to high levels of PAHs and heavy metals. Regardless of development state, urban areas have higher lung cancer rates due to increased exposure to traffic emissions, industrial pollutants, and secondhand smoke. Geographic and socioeconomic factors thus shape pollutant exposure profiles, influencing lung cancer risk globally.

The acceleration of economic growth, urbanization, and industrialization has resulted in the exposure of an increasingly larger population to higher concentrations of PM2.5 (GBD 2019 Diseases and Injuries Collaborators, 2020). PM2.5 exposure is estimated to have risen significantly, increasing by 41.21% from 1990 to 2017 (GBD 2017 Risk Factor Collaborators, 2018). In September 2021, the WHO (2021) revised the annual mean air quality guidelines for PM2.5 from $10\,\mu g/m^3$ to $5\,\mu g/m^3$ because long-term exposure to low concentrations of PM2.5 can also pose considerable health risks. At present, the annual average PM2.5 concentration ranges from $<10\,mg/m^3$ in Canada to $>100\,mg/m^3$ in certain Southeast Asian countries, and the global weighted annual average PM2.5 concentration is approximately $46\,\mu g/m^3$ (Turner et al, 2020).

## Epidemiological studies implicating the role of PM2.5 in lung cancer carcinogenesis

Most lung cancers diagnosed in nonsmokers are non-small cell lung cancers (NSCLCs), predominantly adenocarcinoma (Cho et al, 2017). It is estimated that 15–20% of male and more than 50% of female patients with lung cancer are nonsmokers, especially in East Asia (Lam et al, 2023), suggesting that nonsmoking women have a higher likelihood of developing lung cancer compared to non-smoking men (Parkin et al, 2005). Significant geographic differences exist in the incidence of LCNS. For example, nonsmokers account for approximately 20% of female patients with lung cancer in the United States, whereas the proportion of nonsmokers among female patients with lung cancer in Asia and the Middle East can reach 60–80% (Siegel et al, 2021, 2020). Epidemiologic studies have shown that LCNS occurs more commonly in women and may have a genetic predisposition (Smolle and Pichler, 2019).

In addition to genetic predisposition, another possible cause for LCNS may be exposure to air pollution, such as PM2.5. The study by Myers et al found that never-smokers with lung cancer had higher exposure to PM2.5 compared to ever smokers, highlighting the role of air pollution in lung cancer risk among never-smokers (Myers et al, 2021). The study by Huang et al, using UK Biobank data, found a significant correlation between PM2.5 exposure and increased lung cancer risk, with a hazard ratio (HR) of 1.63 per $5\,\mu g/m^3$ increase in PM2.5 concentration (Huang et al, 2021). Similarly, a recent study using UK Biobank data also revealed that PM2.5 levels were correlated with an increased risk of lung cancer incidence, with a HR of 1.08 per each $1\,\mu g/m^3$ increase in PM2.5 (Hill et al, 2023). A systematic review and meta-analysis revealed that for every $10\,\mu g/m^3$ increase in PM2.5, the relative risk (RR) of lung cancer mortality was 1.12 overall, with variations noted across continents North America (RR = 1.16), Asia (RR = 1.08), and Europe (RR = 1.14) (Zhang et al, 2022). A large nationwide cohort study in the USA revealed that long-term PM2.5 exposure was independently associated with an increased risk of lung cancer incidence. The hazard ratio was 1.008 per $1\,\mu g/m^3$ increase in the incidence of lung cancer (Liu et al, 2023). A case-control study of women with low smoking prevalence in Taiwan showed a dose-response relationship between higher PM2.5 exposure and increased risks of lung adenocarcinoma (Yang et al, 2022). A nationwide case-referent study in Taiwan reported that prolong PM2.5 exposure increased the risk of adenocarcinoma-related lung cancer in both females and males, in people with and without EGFR mutations, and in both early and advanced stages of adenocarcinoma-related lung cancer (Lin et al, 2024). Moreover, PM2.5 levels were linked to higher EGFR-mutant lung cancer in England, South Korea, and Taiwan, with rates rising by 0.63 ($P = 0.0028$), 0.71 ($P = 0.0091$), and 1.82 ($P = 4.01 \times 10^{-6}$) per 100,000 individuals for each $1\,\mu g/m^3$ increase in PM2.5, respectively (Hill et al, 2023). These studies demonstrated a significant correlation between PM2.5 exposure and the occurrence of lung cancer, indicating that PM2.5 exposure can be a factor in the occurrence of LCNS (Bowe et al, 2019; Lo et al, 2022; Sang et al, 2022; Yang et al, 2020b).

Epidemiological studies underscore the role of gene-environment interactions in lung cancer, especially in nonsmokers. Devarakonda et al profiled lung adenocarcinoma in never-smokers, and identified EGFR and TP53 mutations as common drivers (Devarakonda et al, 2021). These mutations often correspond to specific mutational signatures linked to environmental pollutants. Mapping of mutational signatures associated with environmental agents revealed that pollutants like PAHs promote genetic instability through APOBEC cytidine deaminase pathways (Kucab et al, 2019). A proteogenomic study of nonsmoking lung cancer in East Asia linked EGFR and TP53 mutations with PM2.5 and PAH exposure (Chen et al, 2020). These studies emphasize the role of region-specific gene-environment interactions in lung cancer, particularly in nonsmokers. Mutations such as EGFR and TP53, commonly found in never-smokers, are often associated with environmental pollutants like PAHs and PM2.5.

## PM2.5 and EGFR-mediated LCNS

Several prominent studies have reported that air pollution, particularly PM2.5, plays a crucial role in the onset of LCNS (Hamra et al, 2014; Mu et al, 2023; Tseng et al, 2019; Turner et al, 2011). EGFR is a receptor tyrosine kinase that transmits growth factor signals from outside the cell to the cytoplasm. EGFR is a major driver in lung cancer, regulating important tumorigenic processes such as proliferation, apoptosis, angiogenesis, and invasion. EGFR is frequently overexpressed and mutated in NSCLC (Yang et al, 2020a). Genomic studies consistently show that LCNS patients carry a considerably greater frequency of driver gene mutations, particularly those in EGFR, than smokers (Chen et al, 2020; Devarakonda et al, 2021; Zhang et al, 2021). Chen et al (2020) reported an 87% detection rate of EGFR mutations in their LCNS Taiwanese subpopulation. Their studies comparing protein genomic databases of LCNS patients revealed increased expression of proteins associated with PAH or nitro-PAH exposure, particularly in the LCNS of older women with EGFR mutations (Chen et al, 2020). Devarakonda et al identified a diverse ethnic cohort in which 52% of LCNS patients were EGFR mutation-positive. In addition,

84% of LCNS patients in this cohort exhibited a consistent nitro-PAH signature, together with various other PAH mutation patterns (Devarakonda et al, 2021). The nitro-PAH signature and high frequency of EGFR mutations in LCNS suggest that long-term exposure to air pollutants such as PM2.5 may be one of the major causes of LCNS. However, the detailed mechanism by which PM2.5 regulates EGFR oncogenic pathways remains poorly understood. In principle, PM2.5 may be involved in the initiation, selection, and/or promotion of lung cancers. Here, we summarize the latest insights into how PM2.5 contributes to the pathogenesis of LCNS.

First, PM2.5 may be involved in the initiation of lung cancer. Multiple studies have indicated that the key molecular mechanisms underlying PM2.5-induced cytotoxicity are likely related to the induction of DNA damage. DNA damage is considered an early critical event in the initiation of mutations and carcinogenesis (Chu et al, 2015). Personal PM2.5 exposure increased the formation of 7-hydro-8-oxo-2'-deoxyguanosine (8-oxodG) in lymphocyte DNA by 11% per 10 μg/m³ rise in PM2.5 (Sorensen et al, 2003). In addition to producing DNA damages, PM2.5 inhibit the nucleotide excision repair (NER) pathway (Mehta et al, 2008) and downregulate Rad51, a key gene involved in homologous recombination (HR) repair of DNA double-strand breaks (Liu et al, 2020). These disruptions in the HR and NER pathways result in the accumulation of unrepaired DNA lesions and replication errors, thus increase the likelihood of producing mutations and causing genetic instability. KRAS mutations were detected in lung tumors of nonsmokers exposed to PAH-rich coal emission (DeMarini et al, 2001). PM2.5 exposure is also linked to increased lung cancer incidence driven by EGFR and KRAS mutations (Hill et al, 2023). These mutations may impact important regulatory genes such as TP53, a tumor suppressor essential for cell cycle control and apoptosis (Kucab et al, 2019).

Another possible mechanism for the involvement of PM2.5 in LCNS is the selection and promotion of EGFR-mutant lung cancer. A recent study suggested that exposure to ambient air pollution may drive the selective proliferation of preexisting cells with EGFR mutations, offering a potential mechanism for the higher EGFR mutation rate in tumors of individuals residing in high-pollution areas (Hill et al, 2023). Long-term exposure to PM2.5 has been shown to result in persistent EGFR activation, increased cell proliferation, and tumor growth in human adenocarcinoma cells harboring specific EGFR mutations (L858R and T790M). This effect is particularly pronounced in cells with these mutations compared with those with wild-type EGFR. In addition, PM2.5 exposure induces aryl hydrocarbon receptor (AhR)-dependent transcription of transmembrane serine protease 2 (TMPRSS2) and interleukin-18 (IL-18), further promoting cancer progression in EGFR-mutant cells (Wang et al, 2023). These studies suggest that exposure to air pollutants such as PM2.5 can influence the genetic landscape of lung cancer by initiating or/and promoting the growth of cells harboring specific driver mutations, such as those in EGFR.

PM2.5 exposure has been linked to various mechanisms that initiate and promote lung cancer, particularly through the actions of EGFR and AhR (Vogeley et al, 2022; Wang et al, 2023). AhR regulates cancer-related responses through both traditional genomic signaling and nongenomic pathways (Rothhammer and Quintana, 2019). Genomic AhR signaling involves DNA binding to regulate gene expression, while nongenomic AhR signaling activates pathways like EGFR through cytoplasmic protein interactions. AhR can activate EGFR through both traditional genomic signaling and nongenomic pathways, promoting cancer cell proliferation and resistance to EGFR–TKIs by activating proto-oncogene tyrosine-protein kinase Src (Src) signaling (Ye et al, 2018). PM2.5 exposure also activates the Src/STAT3 pathway, resulting in heightened production of vascular endothelial growth factor (VEGF), which further contributes to chronic airway inflammation (Xu et al, 2019; Xu et al, 2016). PM2.5-induced lung inflammation is driven by the elevated ROS to activate pathways like the EGF-EGFR-protein kinase B (AKT)-nuclear factor kappa B (NF-κB) cascade, interleukin-1β (IL-1β), and IL-18 (Jin et al, 2017). PM2.5 exposure enhances NF-κB activity, which in turn drives the production of pro-inflammatory cytokines like interleukin-6 (IL-6) and TNF-α (Liu et al, 2018). In addition, PM2.5 exposure increases IL-1β release from macrophages, encouraging a progenitor-like state in EGFR-mutant alveolar type II cells, which are precursors to lung adenocarcinoma (Hill et al, 2023). PM2.5 exposure also significantly induces IL-18 production to promote lung cancer progression (Wang et al, 2023). Prolonged PM2.5 exposure leads to chronic inflammation, characterized by the ongoing recruitment and activation of immune cells. This chronic inflammatory state creates a microenvironment conducive to DNA damage, cell proliferation, and apoptosis resistance, all of which contribute to tumorigenesis.

PM2.5 not only induces oxidative stress and inflammation but also enhances the prevalence of EGFR mutations while activating oncogenic pathways like Src. The crosstalk between EGFR and Src enhances cancer cell proliferation, survival, and metastasis, while also contributing to resistance against EGFR-targeted therapies. PAHs can induce wild-type EGFR activation via Src kinase triggered by AhR ligand interactions (Vogeley et al, 2022). PM2.5 also activates the EGFR in lung cancer cells, including those with wild-type EGFR (Wang et al, 2023). The cumulative effects of PM2.5 exposure, including ROS generation, inflammation, DNA damage, and activation of oncogenic pathways like EGFR and AhR, create a vicious cycle that promotes lung cancer development and progression. AhR also has the ability to bind to the promoter regions of cMyc (Ouyang et al, 2020), a gene implicated in various stages of lung cancer development, notably in drug resistance. These findings suggest that the interaction between EGFR and AhR extends beyond mere regulation of EGFR signaling and impacts the sensitivity of cancer cells to EGFR–TKIs. The significance of nongenomic AhR signaling in the context of EGFR–AhR crosstalk, particularly in the context of drug resistance, warrants further investigation.

## Clinical implications and future directions

This review presents evidence that PM2.5 exposure can be a major cause of LCNS. Understanding the potential mechanisms by which PM2.5 induces carcinogenesis in LCNS could have important clinical implications. For example, the identification of specific driver mutations in the LCNS should allow clinicians to use the appropriate targeted drugs. The finding that PM2.5 exposure induces inflammation suggests that targeting inflammatory pathways activated by PM2.5 could be an effective strategy to prevent or slow the progression of lung cancer in individuals exposed to high levels of air pollution. Investigating how inflammatory mediators

such as IL-1 and IL-18 drive tumor progression could identify new therapeutic targets (Hill et al, 2023; Wang et al, 2023). Studies using animal models and cellular systems will be invaluable for exploring the intersection of PM2.5 exposure, inflammation, and genetic mutations, leading to the development of anti-inflammatory interventions that complement existing cancer therapies. For example, using CRISPR/Cas9 technology, specific mutations (e.g., in EGFR, TP53) can be introduced into organoids and mice to study the synergy between genetic mutations and PM2.5 induced inflammation in cancer development. Co-culture models that combine lung epithelial cells with immune cells such as macrophages, dendritic cells, or fibroblasts are particularly useful for studying the interaction between immune cells and lung cells in response to PM2.5. Culture of primary human bronchial epithelial cells in a 3D matrix should allow for more physiologically relevant studies of PM2.5 exposure. These in vitro model studies can mimic the airway environment and are useful for assessing inflammatory responses, cytokine production, and genetic mutations.

Detailed studies on nongenomic signaling pathways, particularly the interaction between AhR and EGFR, could provide new insights into how environmental pollutants contribute to the development of resistance to EGFR–TKIs. Understanding these mechanisms is crucial for developing effective treatment strategies for LCNS. For example, combining EGFR–TKIs with agents targeting AhR might overcome resistance and improve patient outcomes.

Large-scale, longitudinal epidemiological studies across diverse populations are essential to confirm current findings and identify additional environmental and genetic risk factors for LCNS. Comparative research between different geographic regions can shed light on variations in LCNS incidence and progression, providing critical data for tailored prevention and screening strategies. These studies should also explore the synergistic effects of PM2.5 with other pollutants and lifestyle factors. PM2.5 often interacts with pollutants like $NO_2$, $SO_2$, and VOCs, which can collectively intensify oxidative stress, inflammation, and DNA damage. In addition, lifestyle factors such as diet, physical activity, and exposure to indoor pollutants (e.g., secondhand smoke, cooking fumes) may exacerbate the impact of PM2.5. For example, inadequate nutrition weakens antioxidant defenses and increases susceptibility to PM2.5-induced oxidative damage.

Effective prevention strategies for high-risk populations, particularly those in heavily polluted areas, are crucial. Expanding lung cancer screening programs to include nonsmokers, particularly those with high PM2.5 exposure or a genetic predisposition, is urgently needed. The development of noninvasive screening tools, such as blood-based biomarkers or advanced imaging techniques, could facilitate early detection and improve outcomes for nonsmokers at risk of lung cancer (Chang et al, 2024). Advanced imaging techniques, like low-dose computed tomography (LDCT), can enhance early lung cancer detection in high-risk populations, including nonsmokers exposed to high levels of PM2.5. Establishing clear follow-up protocols for patients with identified risk factors or early signs of lung cancer can ensure timely intervention. Developing specific guidelines or prediction model for never-smokers that address unique risk factors associated with PM2.5 exposure will enhance targeted screening efforts. Translating findings on PM2.5 exposure, EGFR mutations, inflammation, and

oxidative stress into clinical practice involves creating comprehensive screening and early detection guidelines tailored for non-smokers at risk of lung cancer. By integrating environmental assessments, routine biomarker testing, advanced imaging techniques, public health initiatives, and updated clinical guidelines, healthcare providers can improve early detection rates and outcomes for this vulnerable population (Berg et al, 2023). This approach not only addresses the direct health impacts of air pollution but also enhances overall cancer care strategies in nonsmokers.

In summary, to address the impact of PM2.5 on lung cancer in nonsmokers requires a comprehensive approach that integrates scientific research, public health policy, and clinical practice. By understanding the underlying mechanisms and implementing effective interventions, we can greatly reduce the lung cancer burden and improve outcomes for those affected.

## Conclusion

The intricate relationship between PM2.5 exposure and lung cancer in nonsmokers underscores a significant public health challenge. This review highlights that long-term PM2.5 exposure play a significant role in the incidence and progression of LCNS (Table 1), particularly through mechanisms involving oxidative stress, inflammation, and genetic mutations such as those in the EGFR gene (Fig. 2). These findings emphasize that PM2.5 not only initiates carcinogenesis by causing DNA damage but also promotes the proliferation of preexisting cancerous cells with driver mutations, exacerbating the disease.

Given the established role of PM2.5 in lung cancer, it is crucial to integrate this knowledge into clinical and public health strategies. Identifying and targeting specific genetic mutations and inflammatory pathways activated by PM2.5 can lead to more effective treatment and prevention strategies. For example, combining EGFR–TKIs with agents that inhibit AhR signaling could overcome drug resistance in LCNS patients. Moreover, expanding lung cancer screening programs to include high-risk nonsmokers and developing noninvasive diagnostic tools including artificial intelligence could facilitate early detection and improve treatment outcomes of lung cancer patients.

Upcoming research should focus on comprehensive, long-term studies across diverse populations to further elucidate the environmental and genetic factors contributing to LCNS. In addition, exploring the synergistic effects of PM2.5 with other pollutants and lifestyle factors will provide a more comprehensive understanding of LCNS disease etiology. Comparative studies between different geographic regions can reveal critical insights into regional variations in LCNS incidence and progression, informing tailored prevention and intervention strategies.

Ultimately, to mitigate the impact of PM2.5 on lung cancer in nonsmokers requires a multifaceted approach that encompasses scientific research, clinical practice, and public health policy. By advancing our understanding of the underlying mechanisms and implementing evidence-based interventions, we can significantly reduce the burden of lung cancer and improve the quality of life for those affected by this devastating disease.

**Table 1.   A summary table of key research works on PM2.5 and lung cancer.**

| Focus | Findings | Journal |
|---|---|---|
| Long-term exposure of PM2.5 and risk of lung cancer | Long-term PM2.5 exposure linked to elevated risk of adenocarcinoma lung cancer in Taiwan. | Lin et al (2024). *Environmental Research*, 252:118889 |
| Non-smoking-related lung cancer and PM2.5 | Emerging evidence of lung cancer in never-smokers is partially attributed to PM2.5. | LoPiccolo et al (2024). *Nature Reviews Clinical Oncology*, 21:121–146 |
| LDCT lung cancer screening for never-smokers in Taiwan | Low-dose CT screening among never-smokers in Taiwan reveals potential risk factors tied to PM2.5 exposure. | Chang et al (2024). *Lancet Respiratory Medicine*, 12:141–152 |
| Air pollution elevates risk of lung cancer | A comprehensive review linking air pollution exposure, especially PM2.5, to an elevated risk of lung cancer. | Berg et al (2023). *Journal of Thoracic Oncology*, 18:1277–1289 |
| PM2.5 activates EGFR and promotes lung cancer progression | PM2.5 promotes lung cancer progression via the AhR-TMPRSS2-IL18 pathway. | Wang et al (2023). *EMBO Molecular Medicine*, 15(6):e17014 |
| PM2.5 promotes progression of lung adenocarcinoma | Air pollutants promote lung adenocarcinoma, highlighting the role of PM2.5 in cancer progression. | Hill et al (2023). *Nature*, 616:159–167 |
| Long-term study on PM2.5 and lung cancer in U.S. | Strong association between PM2.5 exposure and lung cancer incidence among older Americans. | Liu et al (2023). *Environmental International*, 181:108266 |
| Lung cancer screening in Asia | Implication of PM2.5 in increased lung cancer incidence in Asia and screening recommendations. | Lam et al (2023). *Journal of Thoracic Oncology*, 18:1303–1322 |
| Oxidative stress from PM2.5 exposure in airway epithelial cells | PM2.5 exposure leads to oxidative stress and inflammatory responses in bronchial epithelial cells. | |
| PM2.5 exposure and risk of Lung adenocarcinoma in women | Higher PM2.5 exposure is associated with an increased risk of lung adenocarcinoma in women. | Yang et al (2022). *Respirology*, 27:951–958 |
| Effects of PM2.5 on lung cancer-related mortality | PM2.5 exposure is linked to lung cancer mortality, but with variations across regions and races. | Zhang et al (2022). *Air Quality, Atmosphere & Health*, 15:1523–1532 |
| Genetic susceptibility of air pollution-related lung cancer | Study using the UK Biobank links genetic susceptibility and PM2.5 exposure to lung cancer risk. | Huang et al (2021). *American Journal of Respiratory and Critical Care Medicine*, 204:817–825 |
| Genomic classification of lung cancer in never-smokers | Classification of lung cancer types in never-smokers indicates PM2.5 as a contributing environmental factor. | Zhang et al (2021). *Nature Genetics*, 53:1348–1359 |
| Air pollution exposure and lung cancer risk in never-smokers vs. ever-smokers | Higher ambient air pollution exposure is linked to lung cancer in never-smokers compared to ever-smokers. | Myers et al (2021). *Journal of Thoracic Oncology*, 16:1850–1858 |
| Molecular signatures of lung cancer in never-smokers | PM2.5 exposure delineates molecular pathways linked to lung cancer progression in East Asia. | Chen et al (2020). *Cell*, 182:226–244.e217 |
| Cancer stem-like properties and PM2.5 exposure | PM2.5 exposure enhances cancer stem-like properties through the aryl hydrocarbon receptor. | Ouyang et al (2020). *Signal Transduction and Targeted Therapy*, 5:78 |
| Lung cancer in nonsmokers and exposure to PM2.5 in Taiwan | Elevated lung cancer risk is observed among nonsmokers exposed to PM2.5. | Tseng et al (2019). *Journal of Thoracic Oncology*, 14:784–792 |
| Mutational signatures from environmental agents in lung cancer | Catalogued mutational signatures indicate a role of PM2.5 as a potent mutagen in lung cancer. | Kucab et al (2019). *Cell*, 177:821–836.e16 |
| PM2.5-related lung cancer mortality in the U.S. | PM2.5 contributes significantly to cause-specific mortality, including lung cancer. | Bowe et al (2019). *JAMA Network Open*, 2:e1915834 |
| PM2.5 and lung inflammation pathways | PM2.5 activates EGF receptor signaling, increasing lung inflammation and potentially leading to cancer. | Jin et al (2017). *Environmental Toxicology*, 32:1121–1134 |
| DNA damage and PM2.5 exposure | PM2.5 exposure increased DNA damage in a Chinese population study. | Chu et al (2015). *Toxicology Letters*, 235:172–178 |
| IARC evaluation on PM2.5 carcinogenicity | PM2.5 is identified as a probable carcinogen based on environmental health data. | Loomis et al (2013). *Lancet Oncology*, 14:1262–1263 |
| PM2.5 and reactive oxygen species (ROS) | PM2.5-induced ROS leads to oxidative stress and inflammation, which may contribute to lung carcinogenesis. | Valavanidis et al (2013). *International Journal of Environmental Research and Public Health*, 10:3886–3907 |
| PM2.5 and lung cancer in nonsmokers | Long-term PM2.5 exposure is significantly associated with lung cancer in nonsmokers. | Turner et al (2011). *American Journal of Respiratory and Critical Care Medicine*, 184:1374–1381 |
| Effects of PM on DNA repair | PM2.5 impedes DNA repair and enhances mutagenic effects in lung tissue. | Mehta et al (2008). *Mutation Research*, 657:116–121 |
| PM2.5 and DNA biomarkers | PM2.5 exposure is linked to DNA damage biomarkers | Sorensen et al (2003). *Cancer Epidemiology, Biomarkers & Prevention*, 12:191–196 |
| PAH-rich PM exposure and lung mutations | KRAS and TP53 mutations in nonsmokers exposed to PAH-rich emissions are linked to PM2.5 exposure. | DeMarini et al (2001). *Cancer Research*, 61:6679–6681 |

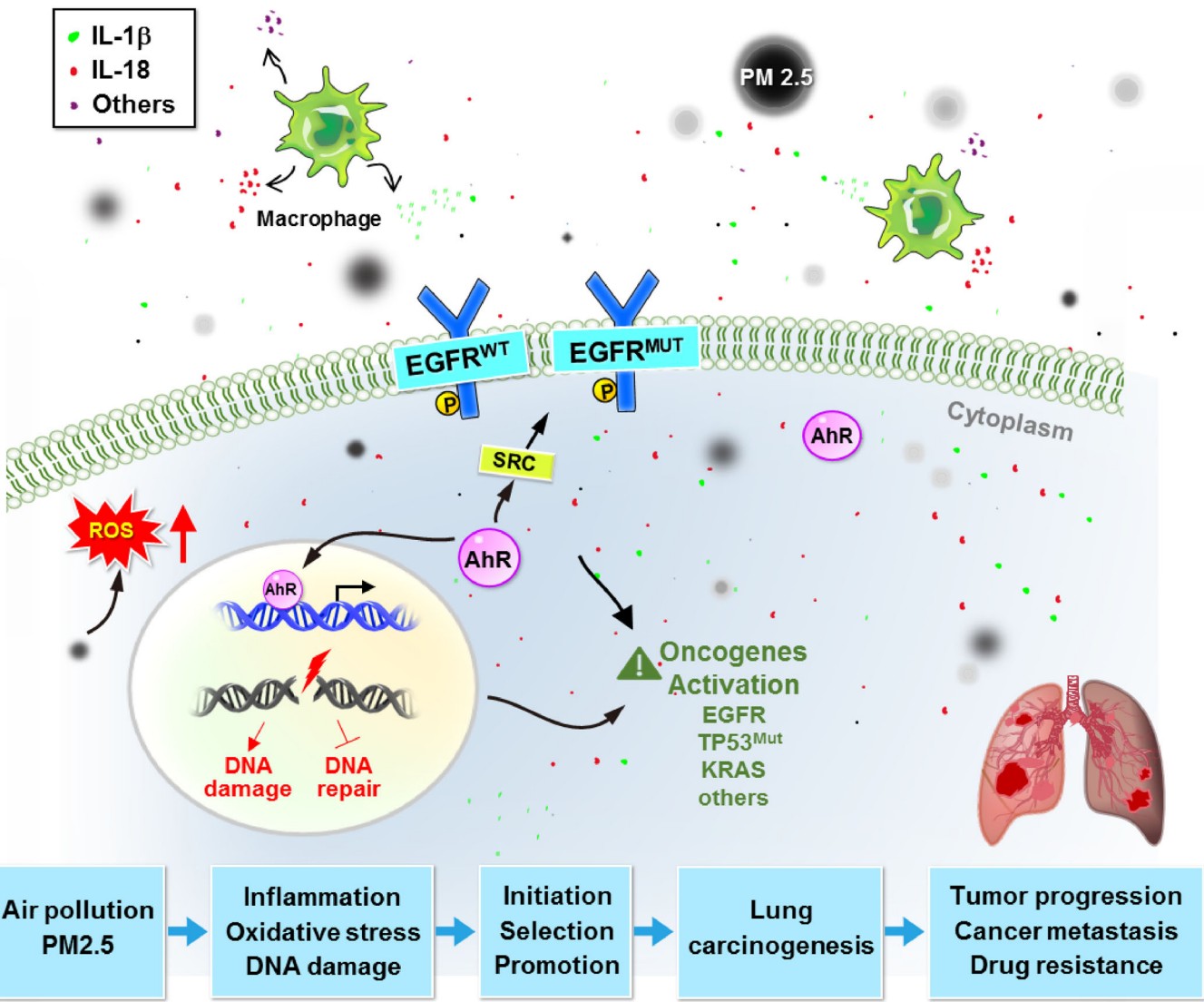

**Figure 2.  PM2.5 may be involved in the initiation, selection, and/or promotion of lung cancer in nonsmokers.**

Schematic showing the molecular pathways by which PM2.5 contributes to lung carcinogenesis. It contributes to lung cancer initiation by inducing inflammation, oxidative stress, and DNA damage. PM2.5 also selects for preexisting EGFR mutations, increasing EGFR activation (wild type and mutant), which leads to oncogene activation, tumor progression, cancer metastasis, drug resistance, and ultimately, lung cancer development. The role of inflammatory cytokines, such as IL-1 and IL-18, is highlighted, indicating their contribution to chronic inflammation and the promotion of tumor progression. In addition, AhR activation by PM2.5 influences EGFR signaling, promoting cancer cell proliferation and resistance to EGFR–TKIs through Src signaling activation.

## Pending issues

### *Clarity of PM2.5 mechanisms*
Further detail is needed on the specific molecular pathways by which PM2.5 induces oxidative stress and inflammation and how these processes directly lead to EGFR mutations and cancer development.

More comprehensive explanations of how PM2.5 exposure promotes the selection and proliferation of cells with EGFR mutations.

### *Detailed epidemiological data*
Incorporation of more recent and diverse epidemiological studies to strengthen the correlation between PM2.5 exposure and LCNS.

Discussion of variations in PM2.5 exposure and LCNS incidence across different geographic regions and populations.

### *Impact of other pollutants*
Examination of the synergistic effects of PM2.5 with other air pollutants, such as nitrogen dioxide and sulfur dioxide, on LCNS.

Analysis of the combined impact of indoor and outdoor air pollution on the risk of lung cancer.

### *Genetic and environmental interactions*
Exploration of how genetic predispositions interact with environmental factors such as PM2.5 to influence the risk of lung cancer in nonsmokers.

Identification of potential biomarkers that could help predict susceptibility to LCNS due to PM2.5 exposure.

### Public health strategies

Development of targeted public health interventions to reduce PM2.5 levels in high-risk areas.

Strategies for educating the public about the risks of air pollution and promoting behaviors to minimize exposure.

### Clinical implications and recommendations

Discussion on how the findings can be translated into clinical practice, including guidelines for screening and early detection of LCNS in nonsmokers.

Potential therapeutic strategies targeting specific molecular pathways activated by PM2.5, such as combining EGFR–TKIs with anti-inflammatory agents.

### Future research directions

Future research priorities, including large-scale longitudinal studies to further elucidate the relationship between PM2.5 and LCNS, are outlined.

## For more information

The corresponding author PC Yang's website: https://x.com/pan20447. https://www.linkedin.com/in/pan-chyr-yang-2a9123291/.

## Peer review information

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

## Acknowledgements

This work was supported by grants from Chang Gung Memorial Hospital (CMRPF1N0041 and CMRPF1N0042 to CY Chen), Chang Gung University of Science and Technology (grant ZRRPF3P0081), and the National Science and Technology Council (NSTC 112-2320-B-255-009-MY3 to CY Chen and NSTC 113-2314-B-002-003- to PC Yang).

## Author contributions

**Chi-Yuan Chen**: Conceptualization; Supervision; Funding acquisition; Writing—original draft; Project administration; Writing—review and editing. **Kuo-Yen Huang**: Conceptualization; Writing—original draft; Writing—review and editing. **Chin-Chuan Chen**: Visualization; Writing—original draft. **Ya-Hsuan Chang**: Visualization; Writing—original draft. **Hsin-Jung Li**: Writing—original draft. **Tong-Hong Wang**: Conceptualization; Visualization; Writing—original draft; Writing—review and editing. **Pan-Chyr Yang**: Conceptualization; Supervision; Funding acquisition; Visualization; Project administration; Writing—review and editing.

## Disclosure and competing interests statement

The authors declare no competing interests.

