## [Peer Review File · EMBO Molecular Medicine]

The Role of PM2.5 Exposure in Lung Cancer: Mechanisms, Genetic Factors, and Clinical Implications

Chi-Yuan Chen, Kuo-Yen Huang, Chin-Chuan Chen, Ya-Hsuan Chang, Hsin-Jung Li, Tong-Hong Wang, and Pan-Chyr Yang

Corresponding authors: Pan-Chyr Yang (pcyang@ntu.edu.tw) , Tong-Hong Wang (cellww@cgmh.org.tw)

Review Timeline:

Submission Date:	7th Sep 24
Editorial Decision:	2nd Oct 24
Revision Received:	30th Oct 24
Accepted:	4th Nov 24

Editor: Zeljko Durdevic

Transaction Report:

2nd Oct 2024

Dear Prof. Yang,

Thank you for the submission of your manuscript to EMBO Molecular Medicine. I am pleased to inform you that we will be able to accept your manuscript pending the following final amendments:

- 1) Please implement referees' suggestions.
- 2) Title: The current title is too general, please consider revising it, e.g. The Role of PM2.5 Exposure in Lung Cancer: Mechanisms, Genetic Factors, and Clinical Implications
- 3) Reduce keywords to max. 5.
- 4) Figures: Please remove labeling "Figure 1" and "Figure 2" from the figures. Implement suggestion from the referee #3 regarding the figure 2. In figure 2, WT and MUT labeling of EGFR should be superscripted (EGFR^{WT} and EGFR^{MUT}), so please remove rhomb shapes and place WT and MUT superscripted in the same box as EGFR.
- 5) Tables: Please add a table summarizing key research works on PM2.5 and lung cancer as suggested by the referee #1.
- 6) If BioRender was used to create the figures, please add following sentence to the figure legends: "Graphics were created with BioRender.com."
- 7) Author contributions: Please remove it from the manuscript and specify author contributions in our submission system. CRediT has replaced the traditional author contributions section because it offers a systematic machine-readable author contributions format that allows for more effective research assessment. Please use the free text boxes beneath each contributing author's name to add specific details on the author's contribution. More information is available in our guide to authors: <https://www.embopress.org/page/journal/17574684/authorguide#authorshipguidelines>
- 8) As part of the EMBO Publications transparent editorial process initiative EMBO Molecular Medicine will publish online a Review Process File (RPF) to accompany accepted manuscripts. This file will be published in conjunction with your paper and will include the anonymous referee reports, your point-by-point response and all pertinent correspondence relating to the manuscript. Let us know whether you agree with the publication of the RPF.

I look forward to receiving the revised version of your manuscript.

Yours sincerely,

Zeljko Durdevic

*** IMPORTANT INFORMATION ***

- 1) a .doc formatted version of the manuscript text (including Figure legends and tables)
- 2) Separate figure files
- 3) a letter INCLUDING the reviewer's reports and your detailed responses to their comments.

Also, and to save some time should your paper be accepted, please read below for additional information regarding some features of our research articles:

- 1) Glossary: EMBO Molecular Medicine articles will be accompanied by a glossary explaining some of the terms used for

laymen. I identified the following:

_____, _____, _____

Could you please help us in identifying terms that may need an "explanation" other terms that we can add to the glossary.

2) For more information: This is a short list of related web links for further consultation by the readers. Could you identify some relevant ones? Examples are patient associations, OMIM related links, databases, authors websites, etc.

3) Pending issues: At the end of each article we will have a box highlighting issues that still need further studies and where research efforts should converge (we call this the Pending issues box). From my reading I would say:

but I can see there may be many more. Could you work on this as well?

4) Disclosure and competing interest statement: Please include a statement declaring any competing commercial interests in relation to your submitted work.

5) Please note that we now mandate that all corresponding authors list an ORCID digital identifier. This takes <90 seconds to complete. We encourage all authors to supply an ORCID identifier, which will be linked to their name for unambiguous name identification.

Currently, our records indicate that the ORCID for your account is 0000-0001-6330-6048.

Link Not Available

-

Thank you,

Zeljko Durdevic

***** Reviewer's comments *****

Referee #1 (Remarks for Author):

This manuscript offers a comprehensive review of the relationship between PM2.5 exposure and lung cancer in nonsmokers (LCNS), emphasizing PM2.5's critical role in the incidence and progression of LCNS through mechanisms such as oxidative stress, inflammation, and genetic mutations, particularly in the EGFR gene. It provides a detailed analysis of epidemiological studies, molecular mechanisms, and clinical implications related to PM2.5 and LCNS. The review spans a wide range of topics, from the fundamental characteristics of air pollution and PM2.5 to the specific mechanisms driving lung cancer development, while also addressing future research directions and public health strategies.

The authors discuss potential therapeutic strategies and emphasize the need for large-scale longitudinal studies to deepen our understanding of the relationship between PM2.5 and LCNS. However, certain areas of the review could benefit from further exploration and improvement:

A more detailed discussion on the specific molecular pathways by which PM2.5 induces oxidative stress and inflammation, particularly how these processes directly contribute to EGFR mutations and cancer development, would be useful.

A table summarizing key research works on PM2.5 and lung cancer development, rather than selective mentions within the text, would enhance clarity.

Given ongoing debates about whether PM2.5 is the primary factor driving lung cancer, the authors could consider discussing other pollutants in their review.

Addressing whether geographic location or a country's development stage influences lung cancer incidence would add another

important dimension to the discussion.

Additional explanations on the correlation between PM2.5, EGFR mutations, and Src activation would provide more depth. When discussing PM2.5's effect on inflammation, it would be beneficial to clarify the role of chronic inflammation in cancer development.

Incorporating more recent and diverse epidemiological studies would strengthen the correlation between PM2.5 exposure and LCNS.

Exploring the synergistic effects of PM2.5 with other pollutants, lifestyle factors, and the interaction between genetic predisposition and PM2.5 exposure could provide a more comprehensive understanding of LCNS etiology.

Further discussion on translating these findings into clinical practice, such as developing specific screening and early detection guidelines for LCNS in non-smokers, would be valuable.

Overall, this manuscript makes a valuable contribution to the field, but addressing these suggestions could enhance its impact and relevance.

Referee #2 (Remarks for Author):

Title: Air Pollution and Lung Cancer

This review summarizes previous studies on air pollution and lung cancer, including the role of PM2.5 in lung cancer carcinogenesis and the possible mechanisms underlying its role in lung cancer in nonsmokers (LCNS). However, several aspects still require further clarification.

1. Lung cancer likely results from both genetic and environmental disturbances in entire network of genes rather than a single gene. However, this review lacks a description of epidemiological research on the role of gene-environment interactions in lung cancer. For example, air pollution has been shown to modify the impact of genetic susceptibility on lung cancer (PMID: 34252012), as participants with high air pollution exposure and high genetic risk had the highest lung cancer risk compared to those with low exposure and low genetic risk. Genetic risk was defined using a polygenic risk score (PRS) based on 18 SNPs reported in the largest lung cancer GWAS of European descent in this study (PMID: 34252012).

2. In Figure 2, the proinflammatory cytokines, such as IL-1 and IL-18, are highlighted. It would be helpful to include some sentences in the main text on how immune cells involved in mediating air pollution-induced inflammation and their effect on lung cancer.

3. For the sentence "Studies using animal models and cellular systems will be invaluable for exploring the intersection of PM2.5 exposure, inflammation, and genetic mutations, leading to the development of anti-inflammatory interventions that complement existing cancer therapies." mentioned in this review, it would be helpful if the authors could provide more details on which animal models or cellular systems, such as lung cells in 3D culture, could potentially be used for this purpose.

4. In this sentence, "A recent study using UK Biobank data revealed that PM2.5 levels were correlated with an increased risk of lung cancer incidence (hazard ratio = 1.08), which is consistent with the results of previous analyses." the author did not provide a citation for "a recent study."

Referee #3 (Remarks for Author):

Peer review of the manuscript EMM-2024-19909 by Chen CY et al. at EMBO Molecular Medicine with the title "Air Pollution and Lung Cancer".

The authors of the manuscript with the number EMM-2024-19909 discuss recent primary literature presenting epidemiological, cellular and molecular evidences of the influence of air pollutants focusing on fine particulate matter with a diameter equal to or less than 2.5 micrometers on the incidence of lung cancer, the leading cause of cancer deaths worldwide. Furthermore, the article is also relevant to the scientific community as it focuses on lung cancer in nonsmokers (LCNS) which already ranks fifth on the list of cancer mortality worldwide. Overall, the Review by Chen CY et al. addresses topics that will become more relevant in the near future due to technological developments related to nanoparticles and the potential consequences of these advances on air pollution. The text of the Review is well structured and in most of the cases well written. However, before I recommend the Review for publication at EMBO Molecular Medicine, I would like to make several suggestions, which should support the authors to exploit better the potential of the topics discussed in the Review, thereby improving their contribution and increasing the impact of the Review to the scientific community.

Major points:

1. The title of the Review is too general, especially considering that the authors focus mainly on the effect of PM2.5 on the incidence of lung cancer. I suggest that the title is changed in order that at least the fine particulate matter (PM2.5) is mentioned.

2. The abbreviation AhR was used without previous explanation. Please, correct. Please check whether other abbreviations need explanation.

3. The authors use a significant portion of the text in the Review to discuss articles proposing the selection and promotion of cells containing EGFR mutations as a mechanism to explain higher incidence of LCNS in areas with higher levels of air pollution. Among these articles are the article by Hill et al., 2023 at Nature, but also the article by the Review authors Wang et al., 2023 at EMBO Mol Med, in which the authors propose that PM2.5 exposure induces AhR-dependent transcription of transmembrane serine protease 2 (TMPRSS2) and interleukin-18 (IL-18) promoting cancer progression in EGFR-mutant cells. While this mechanism is attractive, I suggest that Chen CY et al. elaborate more on other articles that are also mentioned in the Review and that might propose additional mechanisms to explain the effect of PM2.5 on a high incidence of LC. For example:

3.1 The authors should explain more in detail the mechanism proposed by Mehta et al., 2008 and Liu et al., 2020 linking PM2.5 to DNA damage, inhibition of DNA repair and DNA replication errors.

3.2 Further, instead of citing a relatively old Review as Oberdörster G et al., 2007, I recommend substituting this reference by more recent primary literature supporting the role of PM2.5 inducing oxidative stress, cell cycle changes, cell death, lipid peroxidation, and the stimulation of proinflammatory cytokines.

3.3 The authors should explain better in the manuscript the statement "PM2.5 can form stable DNA adducts or promote depurination at damaged nucleotide sites (Sorensen et al, 2003)" commenting on the experimental evidence presented by the authors (Sorensen et al, 2003) supporting this mechanism of action of PM2.5.

3.4 The authors should explain better in the manuscript the statement "During subsequent DNA replication cycles, bulky DNA adducts may be converted into mutations, thereby affecting important regulatory genes such as TP53 (Kucab et al, 2019)" commenting on the experimental evidence presented by the authors (Sorensen et al, 2003) supporting this mechanism of action of PM2.5.

4. Interestingly, in Hill et al., 2023 (cited 6 times in the Review by Chen CY et al) the authors found by ultradeep mutational profiling of histologically normal lung tissue from 295 individuals oncogenic EGFR and KRAS driver mutations in 18% and 53% of healthy tissue samples, respectively. Since EGFR-driven lung cancer is more common in never-smokers, I understand that Hill et al., 2023, as well as the authors of the Review, focus on activation of mutant EGFR by PM2.5 to explain the higher incidence of LCNS in areas with higher levels of air pollution. However:

4.1 Can the authors of the Review find literature supporting a correlation between KRAS mutations and the effect of PM2.5 on lung cancer? In such a case, the Review by Chen CY et al. will increase its impact if the author could comment on this potential correlation by elaborating on potential mechanisms.

4.2 Can the authors of the Review explain the activation of EGFR by PM2.5 in the absence of mutations? I understand that the effect of PM2.5 is stronger on EGFR harboring specific mutations (L858R and T790M). However, some data in the literature suggest that PM2.5 can also activate the wild-type EGFR. Recently, a mechanisms of EGFR activation in SCLC that explains EGFR-TKIs resistance independently of EGFR mutations was published (Rubio K et al., 2023, PMID 37215577). In addition, NSCLC patients undergo histological transformation to SCLC upon acquisition of resistance to the therapy with EGFR-TKIs (Sequist LV et al., 2011; PMID 21430269). A similar mechanism may play role in the activation of wild-type EGFR by long-time exposure to PM2.5. Perhaps the authors could comment on these observations.

5. I suggest that Chen CY et al improve Figure 2 by including the mechanistic aspects listed in points 3. and 4. Above.

6. Chen CY et al did the following statement in the Review:

"These studies suggest that exposure to air pollutants such as PM2.5 can influence the genetic landscape of lung cancer by initiating or/and promoting the growth of cells harboring specific driver mutations."

Please explain in the manuscript what is referred to as "the genetic landscape of lung cancer".

7. Chen CY et al did the following statements in the Review:

"AhR can activate EGFR through both traditional genomic signaling and nongenomic pathways, promoting cancer cell proliferation and resistance to EGFR-TKIs by activating Src signaling (Ye et al, 2018)."

And

"The significance of nongenomic AhR signaling in the context of EGFR-AhR crosstalk, particularly in the context of drug

resistance, warrants further investigation."

Please explain in the manuscript what is referred to as "nongenomic pathways" and/or "nongenomic AhR signaling", ideally with references supporting these expressions.

I hope that my comments support the authors to improve the Review to reach the level of impact of Reviews recently published at EMBO Molecular Medicine.

The authors addressed the remaining editorial issues.

4th Nov 2024

Dear Prof. Yang,

We are pleased to inform you that your manuscript is accepted for publication and is now being sent to our publisher to be included in the next available issue of EMBO Molecular Medicine.

Your manuscript will be processed for publication by EMBO Press. It will be copy edited and you will receive page proofs prior to publication.

You will soon be contacted by our publisher Springer Nature to sign your publishing license. When you login to the customer service website, please use the token/code copied below to waive the article publication charges. Should you experience any difficulty, please email publishing@embo.org.

Waiver token: XXXXXXXXXX
